# Hydrophobic Ionic Liquids for Efficient Extraction of Oil from Produced Water

**Shehzad Liaqat [1], Amir Sada Khan [2], Noor Akbar [3], Taleb H. Ibrahim [1,*], Mustafa I. Khamis [3], Paul Nancarrow [1], Ruqaiyyah Siddiqui [3], Naveed Ahmed Khan [4] and Mohamed Yehia Abouleish [3]**

1    Department of Chemical Engineering, College of Engineering, American University of Sharjah, Sharjah P.O.Box 26666, United Arab Emirates
2    Department of Chemistry, University of Science & Technology, Banuu 28100, Khyber Pakhtunkhwa, Pakistan
3    Department of Biology, Chemistry and Environmental Sciences, American University of Sharjah, Sharjah 26666, United Arab Emirates
4    Department of Clinical Sciences, College of Medicine, Research Institute for Medical and Health Sciences, University of Sharjah, Sharjah 27272, United Arab Emirates
*    Correspondence: italeb@aus.edu

**Abstract:** Produced water contaminated with oil has adverse effects on human health and aquatic life. Providing an efficient method for the removal of oil from produced water is a challenging task. In this study, the effects of carbon chain length and the cation nature of ionic liquids (ILs) on the removal efficiency of oil from produced water were investigated. For this purpose, seven ILs containing the bis (trifluoromethylsulfonyl) imide ($NTf_2$) anion, and various cations such as imidazolium, pyridinium, phosphonium, and ammonium, were employed for the removal of oil from produced water via liquid–liquid extraction. The effects of process parameters such as the initial concentration of oil in produced water, contact time, pH, salinity, phase ratio, and temperature on the removal efficiency of oil were studied and optimized. 1-Decyl-3-methyl-imidazolium bis(trifluoromethylsulfonyl)imide ($[C_{10}mim][NTf_2]$) (IL4) was found to give the highest oil extraction efficiency of 92.8% under optimum conditions. The extraction efficiency was found to increase with increasing cation alkyl chain length from $C_2$ to $C_{10}$. The extraction efficiency of ILs based on cations follows the order imidazolium > ammonium > phosphonium > anpyridinium. Fourier Transform infrared spectroscopy (FTIR) was used to explore the ILs interaction with oil using $[C_{10}mim][NTf_2]$ as a model. In addition, $^1H$ and $^{13}C$ NMR spectra were recorded to obtain a better understanding of the molecular structure of IL and to investigate the peak shifts in H and C atoms. Moreover, the cell viability of the most efficient IL, $[C_{10}mim][NTf_2]$, in human cells was investigated. It has been concluded that this IL exhibited minimal cytotoxic effects at lower concentrations against human cell lines and is effective for the extraction of oil from aqueous media.

**Keywords:** produced water; oil removal; ionic liquids; liquid–liquid extraction

## 1. Introduction

The oil and gas industry continues to play a key role in our daily lives and, as a result, the consumption of oil and gas is increasing [1]. This significant demand for petroleum and its derivatives has increased the extraction processes in the oil and gas industries [2]. The Arabian Gulf produces the largest quantity of petroleum in the world. During the extraction of oil, a huge amount of wastewater is generated, which is known as produced water (PW) [3]. PW represents 80 to 95% of the total liquid waste obtained during the extraction process [4]. It is estimated that almost 75 billion barrels of PW are being generated annually worldwide [5].

Alkaline surfactants and polymers are added for maximum recovery of oil because they lower the interfacial surface tension between the surface of crude oil and water. PW consists of a huge amount of organic and inorganic compounds, including but not limited

to, dissolved and dispersed oil, grease, heavy metal, waxes, chemical, surfactants, salts, microorganisms, and gases [6]. However, its composition and physical and chemical characteristics may vary depending on the geographic location of oil reservoirs, the nature of hydrocarbons produced, operating conditions, and added chemicals. PW is being discharged into oceans and lakes without proper treatment, which has caused serious environmental issues [7,8]. PW has become a major environmental concern due to its complex physiochemical nature, variation in composition, bulk discharged amount, and toxic nature [9]. The contamination of bodies of water by oil is hazardous to human and marine life. Water contaminated with oil and organic compounds has been demonstrated to cause adverse negative effects on human health, soil, underground, and surface water [3,10].

According to a report published by the United Nations World Water Development (UNWD), approximately six billion people will suffer from clean water scarcity by 2050 due to the increasing demand for water, reduction in water sources, and increasing water pollution driven by the dramatic population and economic evolution. It was mentioned that the scarcity of clean water may be even worse by 2050 as the factors of imbalanced growth, accessibility, and demand are being underrated [11]. Currently, the percentage of the global population suffering from water scarcity at least one month per year is reported at 47% [11], as well as 52%, and it is believed it will reach 57% by 2050 [12].

The global demand for clean water is increasing day by day due to economic developments, population growth, and changing consumer behaviors, and it has increased by 600% in the last 100 years [13]. Industrial and domestic water demand is also increasing quickly, along with the highest demand in agriculture [11]. Water pollution has become worse in the last few decades which is associated with population and economic growth [14]. Currently, 12% of the world's population drink from unhealthy sources, and 30% of the population does not have sanitation. In developing countries, 90% of the sewage is discarded into the water without treatment. Industries are discharging about 200–500 megatons of waste into the water each year [14]. These factors; lack of sanitation, the discharging of sewage and industrial waste into the water, along with other effluents, make water more polluted. Other pollutants such as pharmaceutical waste, hormones, personal care product waste, retardants, detergents, caffeine, and nanomaterials are also contaminating the water [15]. In the future, water pollution will increase further due to larger population and economic growth, and a lack of water treatment [16,17]. In brief, the demand for freshwater will increase, and the availability of clean water and water resources will be reduced.

Oil and gas companies, therefore, must treat PW before discharging it into the environment. Multiple techniques have been used for PW treatment [10]. These include physical, chemical, and biochemical methods like gravity separation, hydro-cyclones, membrane separation, filtration electrodialysis, precipitation, and adsorption [18–20]. Most techniques reveal some drawbacks such as high cost, toxic material usage, environmental issues, and a lack of efficiency.

Currently, ionic liquids (ILs) have gained much attention as extractants for the removal of various pollutants from aqueous media. ILs are salt, consisting only of ions, that have very low melting points. Generally, the definition of ionic liquids is that they have a melting point of less than 100 °C [21]. ILs have a low melting point as they consist of bulky and asymmetric ions with a high degree of charge delocalization [22]. As the size of the ion increases, the ion–ion interaction will decrease which prevents the efficient packing of ions in the crystal structure. For applications of ILs in liquid–liquid extraction under ambient conditions, room temperature ionic liquids (RTILs) exhibiting melting points below room temperature are used. RTILs can also be classified as hydrophilic or hydrophobic, depending on their miscibility with water. Hydrophobic ILs have received significant attention in water treatment applications due to their intrinsic properties of negligible vapor pressure, thermal stability, excellent solvation characteristics, and easy isolation from the aqueous stream [23,24]. These properties of ILs make them an environmentally friendly and potentially cost-effective alternative to other toxic solvents with high vapor pressure.

Multiple ILs have been reported for the removal of various pollutants from the aqueous phase via liquid–liquid extraction. The literature described that 1-octyl-3-methylimidazolium hexafluorophosphate ([C$_8$Mim][PF$_6$]) showed excellent extraction abilities to extract polycyclic aromatic hydrocarbons from the aqueous sample using the extraction process [25]. Similarly, ILs also successfully extracted polycyclic aromatic hydrocarbons from sediments, which indicates that ILs can also extract organic compounds from solid matter. A study proposed the use of dispersive liquid phase micro-extraction for increased extraction efficiencies instead of temperature-dependent extraction of pyrethroid pesticides using [C$_6$Mim][PF$_6$] [26]. The removal of organic sulfur from fuels with six different ILs also has been explained in the literature [27]. The extraction of uranium using ILs in liquid–liquid extraction and the demonstrated feasibility of using ILs for metal ion and uranium separation have also been studied [28]. Liquid–liquid extraction of toluene from toluene/heptane mixtures has also been reported by using multiple ILs, which also highlights the extraction of aromatic hydrocarbons from a mixture of aromatic and aliphatic compounds [29]. The practice of ILs instead of organic solvents for lanthanide extraction in industrial LLE has been examined [30]. Imidazolium-based ILs have been reported for the removal of polyunsaturated fatty acids methyl esters from a variety of alkanes [31]. The application of ILs in hollow-fiber-based liquid phase micro-extraction can eliminate the use of organic solvents for lead and nickel determination in biological and environmental samples [32]. A research paper reported the use of quaternary ammonium-based ILs for the extraction of aromatic amines and phenols [33]. So, different studies have successfully applied ILs for the extraction of different organic compounds which indicates that the IL can be tested for the removal of oil from PW.

In this study, seven different [NTf$_2$]-based hydrophobic ILs with different cations were employed for the removal of oil from PW using liquid–liquid extraction at room temperature and pressure. To the best of our knowledge, these ILs have not previously been reported for the removal of oil from PW. The effect of the various cation with different alkyl chain lengths on the extraction efficiency was studied. The effects of process conditions such as the initial concentration, contact time, phase ratio, pH, salinity, and temperature on the removal efficiency of oil from PW by the selected ILs were also investigated. The toxicity analysis of the most efficient ILs was also performed. The experimental results showed that our best IL is non-toxic and can remove 92.8% of the oil from the PW.

## 2. Materials and Instrumentation

### 2.1. Materials

Heavy crude oil (HCO) was provided by Abu Dhabi National Oil Company (ADNOC) (Abu Dhabi, United Arab Emirates). The surfactant (ENDOR OCC9783) was received from Suez Company (Dubai, United Arab Emirates). ILs were received from Sigma Aldrich and Iolitec, Germany. All other chemicals were of analytical grade and were used without further purification. Synthetic PW is prepared by mixing heavy oil with a specific ratio of the deionized water-surfactant mixture (60 (W):40 (S)). Double distilled water was used throughout all experiments (Water Still Aquatron A4000D, ELGA Lab Water, Lane End, Buckinghamshire, UK). Sodium chloride (NaCl) was used to study the salt effect.

### 2.2. Instrumentation

The concentration of oil in different samples was determined spectrophotometrically using a UV-Vis spectrophotometer (Thermo Fisher Scientific, Model:Evolution 220, Shanghai, China). Fourier Transform infrared spectrophotometer (FTIR, Perkin Elmer, Waltham, MA, USA) was used to determine the presence of the functional group in ionic liquids before and after oil removal. Nuclear Magnetic Resonance (NMR) analysis of the IL4 was performed with Burker Avanve III HD (500 MHZ magnet) with standard Bruker pulse for 1H and 13C experiments. Stuart Vortex mixer (UK) was used for the mixing of samples. Oil and water layers were separated using a centrifuge (HERMLE Labortechnik, Wehingen, Germany). A pH meter (Okton pH 510 series, manufactured by EUTECH INSTRUMENTS, klang, Selangor, Malaysia) was used to measure the pH of the solutions. To adjust the pH

of all solutions, 0.1 M HCl and 0.1 M NaOH were used. The thermophysical properties of the selected ILs, along with their purity, are listed in Table 1. The structure of cations and anions of tested ILs, with their abbreviations, are shown in Table 2.

**Table 1.** Thermophysical properties of the ILs tested in this work.

| No. | Name | Abbreviations | Molecular Formula | Molar Mass (g/mol) |
|---|---|---|---|---|
| IL1 | 1-Ethyl-3-methylimidazolium bis (trifluoromethylsulfonyl) imide | [C$_2$Mim] [NTf$_2$] | C$_8$H$_{11}$F$_6$N$_3$O$_4$S$_2$ | 391.30 |
| IL2 | 1-Butyl-2-3-dimethylimidazolium bis (trifluoromethylsulfonyl) imide | [C$_4$Mim] [NTf$_2$] | C$_{11}$H$_{17}$F$_6$N$_3$O$_4$S$_2$ | 433.39 |
| IL3 | 1-Methyl-3-octyl-imidazolium bis (trifluoromethylsulfonyl) imide | [C$_8$Mim] [NTf$_2$] | C$_{14}$H$_{23}$F$_6$N$_3$O$_4$S$_2$ | 475.47 |
| IL4 | 1-Decyl-3-methylimidazolium bis (trifluoromethylsulfonyl) imide | [C$_{10}$Mim] [NTf$_2$] | C$_{16}$H$_{27}$F$_6$N$_3$O$_4$S$_2$ | 503.50 |
| IL5 | 1-Butyl-4-methylpyridinium bis (trifluoromethylsulfonyl) imide | [C$_4$Mpy] [NTf$_2$] | C$_{12}$H$_{16}$F$_6$N$_2$O$_4$S$_2$ | 430.39 |
| IL6 | Tributyl methyl phosphonium bis (trifluoromethylsulfonyl) imide | [P$_{1,4,4,4}$] [NTf$_2$] | C$_{15}$H$_{30}$F$_6$NO$_4$PS$_2$ | 497.50 |
| IL7 | Butyl trimethylammonium bis (trifluoromethylsulfonyl) imide | [N$_{1,1,1,4}$] [NTf$_2$] | C$_9$H$_{18}$F$_6$N$_2$O$_4$S$_2$ | 396.37 |

**Table 2.** Chemical Structure of the cations and the imide anion of the ILs used in this work with their abbreviations.

| No. | Ionic Liquids | Cations | Anions |
|---|---|---|---|
| IL1 | [C$_2$Mim] [NTf$_2$] |  |  |
| IL2 | [C$_4$Mim] [NTf$_2$] |  | |

**Table 2.** *Cont.*

| No. | Ionic Liquids | Cations | Anions |
|---|---|---|---|
| IL3 | [C$_8$Mim] [NTf$_2$] | $CH_2(CH_2)_6CH_3$ (imidazolium ring with N–$CH_3$) | |
| IL4 | [C$_{10}$Mim] [NTf$_2$] | $CH_2(CH_2)_8CH_3$ (imidazolium ring with N–$CH_3$) | |
| IL5 | [C$_4$Mpy] [NTf$_2$] | $CH_3$ (pyridinium ring) $(CH_2)_3CH_3$ | |
| IL6 | [P$_{1,4,4,4}$] [NTf$_2$] | $(CH_2)_3CH_3$ $CH_3-P^+-(CH_2)_3CH_3$ $(CH_2)_3CH_3$ | |
| IL7 | [N$_{1,1,1,4}$] [NTf$_2$] | $CH_2(CH_2)_2CH_3$ $CH_3-N^+-CH_3$ $CH_3$ | |

## 3. Methods

### 3.1. Preparation of Synthetic Produced Water

The water–surfactant solution with a water–surfactant ratio of 60:40 was prepared by mixing 40 mL of surfactant with 60 mL of de-ionized water [34]. The solution was then sonicated for 10 min to achieve a homogeneous solution. The produced water solutions of different concentrations were prepared by adding a specific mass of oil to the surfactant–water solution and sonicated for 10 min [35]. The concentration range of oil in surfactant–water solution was in the range of 100–700 mg/L.

### 3.2. Liquid–Liquid Extraction

The removal of oil from produced water by various ILs was performed using selected liquid–liquid extraction. A known amount of IL was added to a certain volume of synthesized produced water followed by vortex mixing at a speed of 2500 rpm for specific minutes. The heterogeneous solution was then centrifuged at 3500 rpm for 10 min. The water layer was separated, and its oil content was determined by extraction with hexane and quantified spectrophotometrically at λ = 275 nm by a suitable calibration curve of oil in hexane. The calibration curve of oil in hexane ranges from 3.13 mg/L to 200 mg/L, with a detection limit of 2 mg/L in hexane with $R^2$ = 1 at λ = 275 nm [35]. For this purpose, 1 mL of the water layer was extracted with 9 mL hexane. Experiments were repeated 3 times for each parameter study, and the average of those values was taken. In addition, the standard deviation and uncertainties (σ) of the readings were also recorded for each parameter. The removal efficiency was determined by using Equation (1).

$$R = \left( \frac{C_i - C_f}{C_i} \right) \times 100 \tag{1}$$

where $R$ represents the removal efficiency of ionic liquids, $C_i$ and $C_f$ represent the concentration of oil in produced water in mg/L before and after treatment.

### 3.3. Single Parameter Optimization Process

Oil extraction from produced water using ILs depends on different parameters. Therefore, it is necessary to study the effect of these parameters. In this study, the effects of different initial concentrations of produced water (100–600 mg/L), contact time (2–10 min), pH (2–12), temperature (25–65 °C), salinity (0–2000 mg/L), and phase ratio (PW:IL 1–8) on the removal efficiency of IL were studied, and process conditions were optimized to achieve the maximum removal efficiency. The section below represents the optimized process conditions.

#### 3.3.1. Effect of the Initial Concentration of Oil in PW

To investigate the effect of the initial concentration of oil in PW, solutions of different concentrations of oil in PW ranging from 200 mg/L to 600 mg/L were prepared by using a calculated amount of oil. Following that, IL and PW were mixed with a volumetric ratio of 1:5 mL for each concentration and mixed using a vortex mixer at 2500 rpm for 5 min at room temperature. At the end of each experiment, the remaining concentration of oil in produced water was determined for all concentrations.

#### 3.3.2. Effect of Contact Time

To investigate the effect of contact time on removal efficiency of IL for oil extraction from PW, the experiments were performed at different times ranging from 2 min to 10 min using an initial concentration of 500 mg/L, the volumetric phase ratio of PW:IL was 5:1 mL at room temperature.

### 3.3.3. Effect of pH

The effect of pH for oil extraction from produced water using hydrophobic ILs was studied by using solutions of different pH ranging from 2 to 12 using the initial concentration of 500 mg/L at the optimized contact time with a volumetric phase ratio of PW:IL of 5:1 mL.

### 3.3.4. Effect of Phase Ratio

To study the effect of the phase ratio of IL to produced water for oil removal from PW, experiments were performed at different phase ratios of PW:IL ranging from 1:1 mL to 8:1 mL at the optimized values of the initial concentration, contact time, and pH.

### 3.3.5. Effect of Temperature

To study the effect of temperature on oil extraction using ILs, the experiments were performed at different temperatures ranging from 25 °C to 65 °C at the optimized values of the initial concentration, contact time, pH, and phase ratio.

### 3.3.6. Effect of Salinity

To study the effect of salt on oil removal from PW, solutions of PW having different concentrations of salt ranging from 250 to 2000 mg/L were prepared. For each solution, the required amount of Il was added, and the oil is extracted at the optimized condition of initial concentration, contact time, pH, phase ratio, and temperature.

### 3.3.7. Cell Viability Assays

For the cellular viability, 3-(4,5-dimethylthiazol-2-yl)-2, 5-diphenyl-2H-tetrazolium bromide (MTT) assay was performed as described elsewhere [36]. Briefly, human cells were grown up to confluency, and ILs were incubated with cell monolayer at different concentrations of 0.5, 0.1, 1.5, and 2 mM with 5% $CO_2$ and 95% humidity at 37 °C for 24 h. After this, 20 μL of freshly prepared MTT solution dye was added to each well plate. The plate was incubated for 3–4 h at 37 °C. Next, 100 μL dimethyl sulfoxide (DMSO) was added to each well to dissolve the formazan crystals made by viable cells. DMSO alone was taken as the negative control while 1-butyl-1-methylpyrrolidinium bis-(trifluoromethylsulfonyl)imide, a commercial IL, was taken as the positive control. Percent cell viability was assessed as follows; Percent cell (% Cell) was calculated using Equation (2).

$$\% \text{ Cell viability} = \left( \frac{\text{Test sample}_{\text{Mean OD}}}{\text{Test sample}_{\text{Mean OD}}} \right) \times 100 \tag{2}$$

## 4. Results and Discussion

### 4.1. Screening of ILs

In this work, initially, seven [NTf$_2$]-based hydrophobic ILs having different cations, such as imidazolium, pyridinium, ammonium, and phosphonium, were employed for the efficient removal of oil from PW by liquid–liquid extraction at the initial concentration = 500 mg/L, contact time = 5 min, phase ratio (IL:PW) = 1:5, and temperature = 25 °C. The effect of the molecular structure of the ILs such as cation nature and alkyl chain length on the extraction efficiency was studied to obtain a fundamental understanding of the ILs' behaviors toward the extraction process. To investigate the effect of cation alkyl chain length of ILs on the removal efficiency of oil, imidazolium-based ILs of various alkyl chain lengths ranging from $C_2$ to $C_{10}$ were tested. Table 3 represents the extraction efficiency of examined ILs toward oil removal from produced water. An inspection of Table 3 reveals that the imidazolium-based ILs(IL1-IL4) showed an increasing trend in extraction efficiency with the following order of IL1(80%) < IL2(82%) < IL3(83%) < IL4(85%). This increasing trend in the extraction efficiency for the imidazolium-based ILs is due to the increase in alkyl chain length of the imidazolium- cation-based ILs ranging from $C_2$(IL1) to $C_{10}$(IL4). The main driving force for the extraction process is the hydrophobic interaction between the cation of IL and oil and the hydrogen bonding between the anion of IL and oil [37–39]. As the cation alkyl chain length

increases, the hydrophobicity of the ILs increases. This increased hydrophobicity resulted in the increased hydrophobicity bonding between the cation of the IL and oil which enhances the extraction process [37]. It is also evident from Table 3 that cation nature also has a significant effect on extraction efficiency. The maximum removal efficiency of (85%) was recorded for imidazolium cation whereas the pyridinium-based IL showed minimum removal efficiency (63%). The result in Table 3 shows that removal efficiencies follow the overall pattern of IL4 > IL3 > IL2 > IL1 > IL7 > IL6 > IL5. The results confirmed that the structure of IL has a considerable effect on the removal of oil from PW. It has been documented that the structure of ILs affects the removal efficiency of different pollutants from the aqueous phase [40–42].

**Table 3.** Screening of the seven ILs for the extraction of oil from PW at the initial concentration = 500 mg/L, contact time = 5 min, phase ratio (IL:PW) = 1:5, and temperature = 25 °C.

| Ionic Liquid No. | Ionic Liquids Name | Removal Efficiency (%) |
|:---:|:---:|:---:|
| IL1 | 1-ethyl-3-methylimidazolium bis (trifluoromethylsulfonyl) imide | 80 |
| IL2 | 1-Butyl-2-3-dimethylimidazolium bis (trifluoromethylsulfonyl) imide | 82 |
| IL3 | 1-methyl-3-octyl-imidazolium bis (trifluoromethylsulfonyl) imide | 83 |
| IL4 | 1-Decyl-3-methylimidazolium bis (trifluoromethylsulfonyl) imide | 85 |
| IL5 | 1-Butyl-4-methyl pyridinium bis (trifluoromethylsulfonyl) imide | 63 |
| IL6 | Tributyl methylphosphonium bis (trifluoromethylsulfonyl) imide | 68 |
| IL7 | Butyl trimethylammonium bis (trifluoromethyl sulfonyl) imide | 70 |

### 4.2. Effect of Alkyl Chain Length of Cations

To study the effect of the alkyl chain length of cations, Imidazolium-based ILs with different carbon chain lengths $C_2$Mim(IL1), $C_4$Mim(IL2), $C_8$Mim(IL3), and $C_{10}$Mim(IL4) were selected for such purpose since they contain the same anion with cations differing only in their chain length. The results in Figure 1a show the effect of the carbon chain length on the efficiency of liquid-liquid extraction of oil from PW. Inspection of this figure reveals that the removal efficiency is increasing with the increase in the carbon chain length. The removal efficiency of 80%, 82%, 83%, and 85% was recorded for the carbon chain length of $C_2$Mim(IL1), $C_4$Mim(IL2), $C_8$Mim(IL3), and $C_{10}$Mim(IL4), respectively. IL4, [$C_{10}$Mim][NTf$_2$], showed a maximum efficiency of 85%. As alkyl chain length increases, the interfacial surface tension (IFT) between oil and water decreases which leads to an increase in mass transfer between the two phases which results in increasing the extraction efficiency of oil from PW. Short alkyl chain lengths have lower mass transfer ability compared to higher alkyl chain lengths [43]. It was reported that the hydrophobicity increases with increasing alkyl chain length and thus leads to an increase in mass transfer between the two phases [44]. Furthermore, a study by Zhu et al. showed that imidazolium-based ionic liquids with NTf$_2$ anions displayed lower extraction efficiency at lower alkyl chain lengths [42].

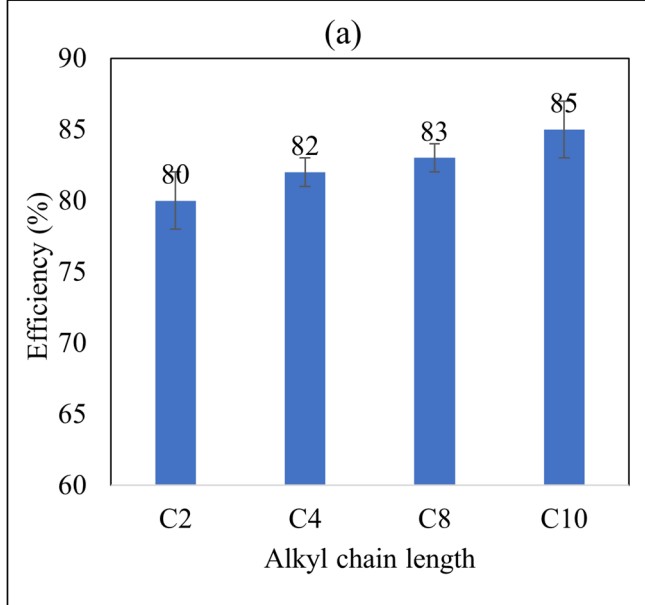
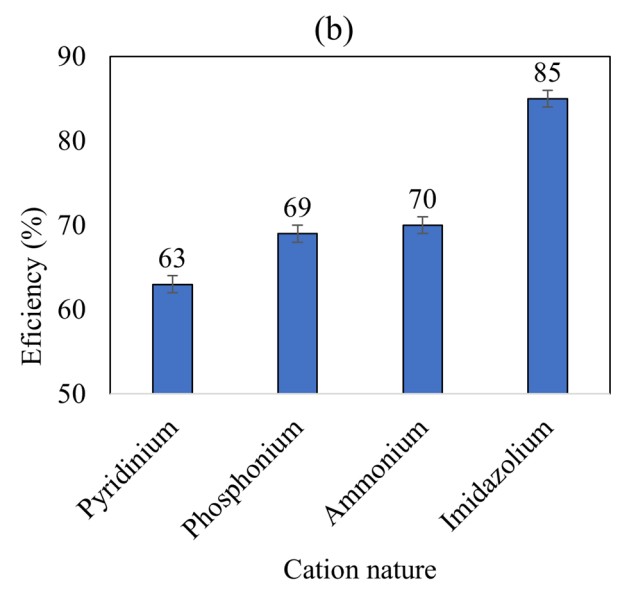

**Figure 1.** (**a**) Effect of alkyl chain length on the removal efficiency of oil from PW using different ILs at initial concentration = 500 mg/L, contact time = 5 min, phase ratio (IL:PW) = 1:5, and temperature = 25 °C, with σ = ±4.02 × 10⁻¹%, (**b**) Effect of cation nature on the removal efficiency of oil from PW using different ILs with the same anion at initial concentration = 500 mg/L, contact time = 5 min, phase ratio (IL:PW) = 1:5, and temperature = 25 °C, with σ = ±3.33 × 10⁻¹%.

### 4.3. Effect of the Nature of the Cation

The nature of the cation in ILs affects the extraction efficiency of different pollutants. To study the effect of cation nature, different ILs having various cations such as imidazolium, pyridinium, phosphonium, and ammonium with the same anion were studied. The results in Figure 1b show that the ionic liquid with imidazolium cation is the best among all these ILs for oil removal from PW.

The removal efficiency of oil from PW by these ionic liquids follow the order: imidazolium (85%) > Ammonium (70%) > Phosphonium (69%) > Pyridinium (63%). This could be attributed to the effect of charge delocalization as manifested by the electronegativity of the central cation on the coulombic attraction with the counter anion which leads to different hydrophobicity of the ILs. This interaction will presumably affect the mass transfer of the oil between the two phases. Our results are also parallel with a previous report that imidazolium-based Ntf₂ ILs with long alkyl chain lengths are successful in separating oil–water emulsion [43]. Furthermore, it was reported that imidazolium-based ILs are efficient in the removal of pollutants from contaminated water using liquid–liquid extraction [42].

### 4.4. Effect of the Initial Concentration of Oil in PW

To investigate the effect of the initial concentration of oil in PW on its removal efficiency, IL4 was selected since it gave the highest removal efficiency under optimized conditions. Figure 2a displays the removal efficiency of oil from synthetic PW using IL4 as a function of the initial oil concentration. Inspection of this figure reveals that the removal efficiency increases with an increasing initial concentration of oil in PW. This increase in extraction efficiency could be attributed to the fact that as the concentration increases, the distribution equilibrium will shift to the IL phase according to Leshatlier's principle, thus leading to the observed increase in oil removal by the IL. Our results are parallel with the literature which showed that when imidazolium-based NTf₂ ILs were used for the extraction of phenol from the aqueous phase, the extraction efficiency of these ILs increases with an increase in concentration [43].

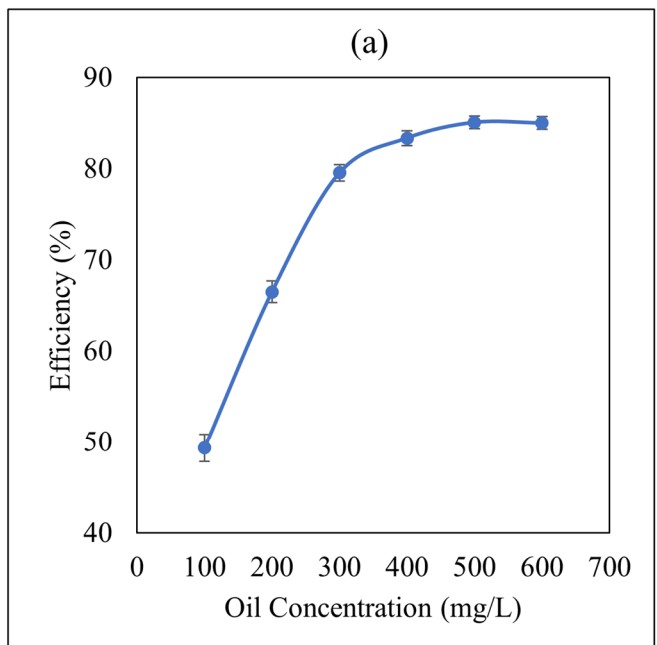 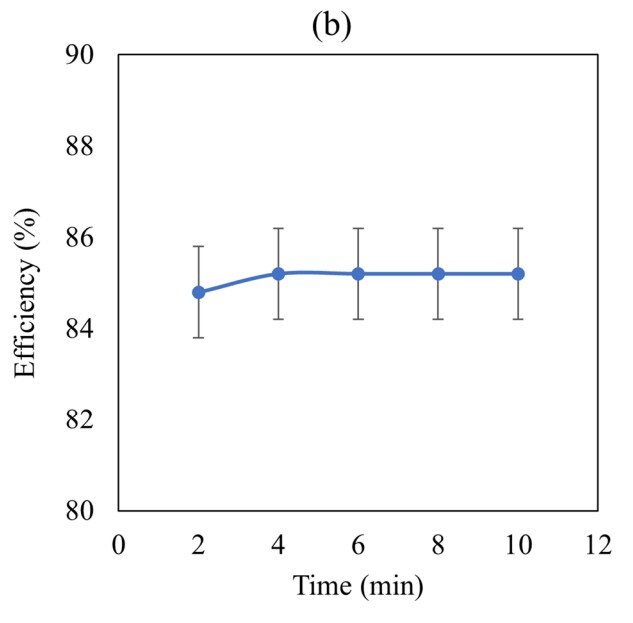

**Figure 2.** (**a**) Effect of the initial concentration of oil in PW on its removal efficiency by IL4 at Phase ratio (IL:PW) = 1:5, contact time = 5 min, and temperature = 25 °C, with σ = ±5.16 × 10⁻¹%, (**b**) Effect of the contact time on the removal efficiency of oil by IL4. Initial concentration of oil 500 mg/L, phase ratio (IL:PW) = 1:5, temperature = 25 °C, with σ = ±2.68 × 10⁻¹%.

### 4.5. Effect of Contact Time

Contact time between PW and IL is an important factor for the liquid–liquid extraction of oil from PW because it gives us information about the equilibrium limit of mass transfer that the maximum mass transfer of the oil from PW to IL has occurred and equilibrium has been achieved [38]. Therefore, it is necessary to know the optimal contact time required for the efficient removal of oil from PW under a given set of conditions. Figure 2b displays the results using IL4 and shows that the removal process is very fast and maximum mass transfer has been achieved within the initial 2 min with an extraction efficiency of 84.8% and equilibrium mass transfer has been achieved at 4 min with a maximum extraction efficiency of 85.2% with no significant effect of higher contact time. ILs which show good removal efficiency in a short time are most suitable and cost-effective for industrial applications. An equilibrium contact time of 4 min was selected as the optimum contact time for further study because extraction efficiency becomes constant after this time due to the equilibrium limit. A similar trend has been reported for the effect of contact time on removal efficiency for wastewater treatment using hydrophobic ILs [38].

To evaluate the applicability of ILs in industrial applications, a comparison study between the contact time obtained in this work and the contact time reported for other adsorbents using solid phase extraction is reported. Table 4 summarizes the results. It is clear from Table 4, that the time required to reach equilibrium in this study is far less than any of the adsorbents reported in the literature without compromising the removal efficiency.

**Table 4.** A comparison between the reported equilibrium time and efficiency of the extraction of oil by different adsorbents using solid phase extraction and IL4 (this study) using liquid–liquid extraction.

| No. | Material | Efficiency (%) | Contact Time (min) | Reference |
|-----|----------|----------------|--------------------|-----------|
| 1 | olive leaves | 80 | 80 | [35] |
| 2 | pomegranate peel | 92 | 50 | [45] |
| 3 | multiwalled carbon nanotubes and their derivates | 85 | 20 | [46] |
| 4 | Graphene nanoplatelets | 90 | 60 | [34] |
| 5 | graphene magnetite | 72.20 | 30 | [34] |
| 6 | eggplant peels | 95 | 40 | [47] |
| 7 | IL 4 (at a phase ratio of 1:1) | 92.8 | 4 | This study |

### 4.6. Effect of pH

The pH of the solution is an important factor during liquid–liquid extraction and is known to affect the removal efficiency of oil from PW. Therefore, the effect of pH on oil extraction from PW using IL4 was studied. For this purpose, PW solutions of different pH range from 2 to 12 were investigated. Inspection of Figure 3a reveals that the removal efficiency is increasing with increasing pH until pH 8 where a plateau is observed. So, a pH value of 8 was selected as an optimal pH for further study. The slightly basic region is the most favorable for the removal of oil from PW. It is well known that the pH affects the speciation of pollutants in oil, and this affects their distribution to the IL phase. At low pH, it is expected that pollutants exist in the molecular form whereas at high pH they undergo ionization giving charged ions that have stronger attraction and affinity towards the cations and anions of the IL, thus leading to the observed increase in the removal efficiency as a function of pH. The plateau at pH 8, indicates that maximum ionization was reached with no further increase in removal efficiency.

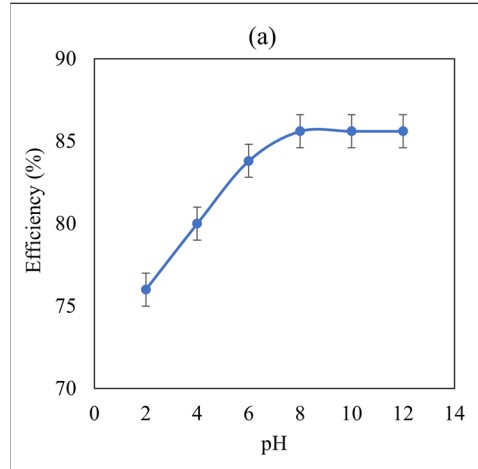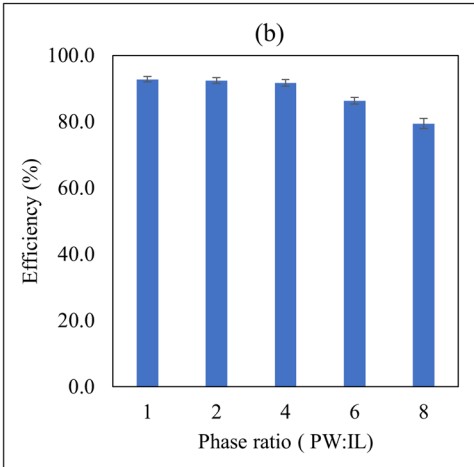

**Figure 3.** (**a**) Effect of the pH on the removal efficiency of oil by IL4, Initial concentration of oil 500 mg/L, phase ratio (IL:PW) = 1:5, temperature = 25 °C, and contact time = 4 min, with $\sigma = \pm 3.09 \times 10^{-1}\%$, (**b**) Effect of the phase ratio on removal efficiency of oil from PW by IL4. Initial oil concentration = 500 mg/L, contact time = 4 min, pH = 8 and temperature = 25 °C, with $\sigma = \pm 3.64 \times 10^{-1}\%$.

### 4.7. Effect of Phase Ratio

The phase ratio is an important factor in liquid–liquid extraction, and it has significant effects on the removal efficiency of oil by ILs. Therefore, the effect of the phase ratio (PW: IL)

was studied. For this purpose, IL4 was used with different values of phase ratios ranging from 1:1 to 8:1 and an initial oil concentration of 500 mg/L, contact time of 4 min, and pH = 8 (Figure 3b). Inspection of Figure 3b reveals that the removal efficiency of oil from PW by IL4 decreases with an increase in the phase ratio. It is evident from Figure 3b that the maximum efficiency (92.8%) was achieved at a phase ratio of 1:1. As the phase ratio changes from 1:1 to 8:1, the removal efficiency has decreased significantly from 92.8% to 79.4%. This decrease in removal efficiency could be attributed to the fact that at a higher phase ratio, many molecules of oil are present in PW to be extracted by the IL which leads to its saturation. Similar behaviors for the removal of pollutants from the aqueous phase using ILs were reported in the literature [38]. A phase ratio of 4:1 was used for the further experiment to save the amount of IL because there is no significant change in extraction efficiency in phase ratios of 1:1 and 4:1.

### 4.8. Effect of Temperature

A temperature study is helpful for the determination of a suitable temperature range where the system can be applied. It also gives information about the exothermic or endothermic nature of the system. A thermodynamic study of the system can also be performed with the help of a temperature study. Temperature can affect the removal efficiency of oil from produced water. It is highly recommended that an extraction process that leads to high efficiency at a wide temperature range is more desirable for industrial applications. Therefore, in this work, the effect of temperature on the extraction efficiency of oil from PW using IL4 was studied. For this purpose, experiments were performed at different temperatures ranging from 25 °C to 65 °C at the optimum conditions of initial concentration of 500 mg/L, contact time of 4 min, pH = 8, and phase ratio = 4:1 (Figure 4a). Inspection of Figure 4a reveals that the extraction efficiency was slightly increased with the increase in temperature. This observation indicates that the extraction process is endothermic. Overall, there is no significant effect of temperature on the extraction efficiency of oil by IL4, hence, ambient temperature can be used to save energy and cost. A similar trend was reported in the literature for wastewater treatment using ILs [38]. Similar results were reported by Lakshmi et al. for the extraction of phenol and its derivates [39,48].

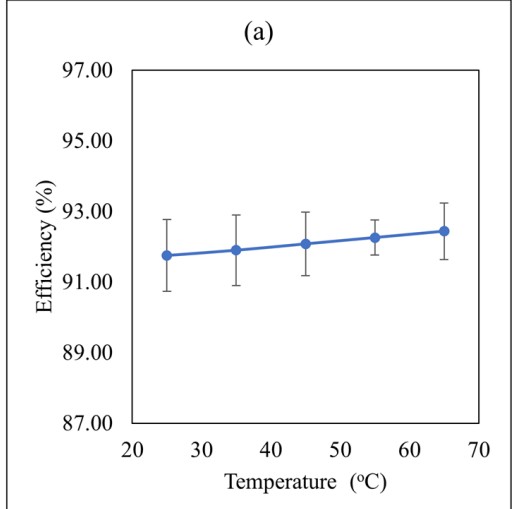
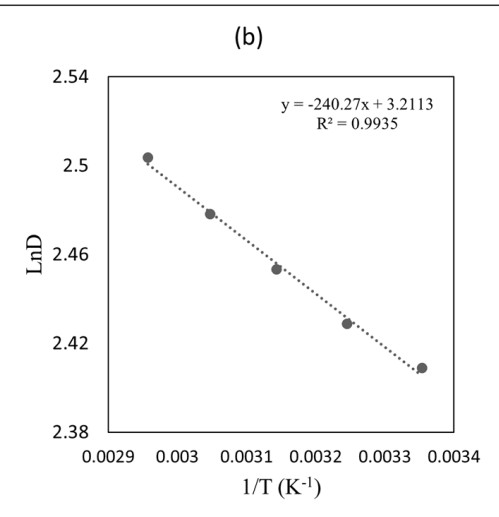

**Figure 4.** (**a**) Effect of the temperature on removal efficiency of oil by IL4. Initial oil concentration = 500 mg/L, contact time = 4 min, pH = 8 phase ratio (PW:IL) = 4:1, with $\sigma = \pm 3.24 \times 10^{-1}$%, (**b**) Van't Hoff plot for oil extraction from PW using IL4. Initial oil concentration = 500 mg/L, contact time = 4 min, pH = 8 phase ratio (PW:IL) = 4:1 and temperature = 25 °C.

### 4.9. Thermodynamic Study of Oil Extraction by ILs

The distribution of oil between the aqueous phase and IL phase can be represented by Equation (3).

$$Oil(aq) \Leftrightarrow Oil(IL) \tag{3}$$

The equilibrium distribution coefficient ($D$) is given in Equation (4).

$$D = \left( \frac{[mass\ of\ Oil]_{IL}}{[mass\ of\ Oil]_{aq}} \right) \tag{4}$$

Equation (5) is used for calculating its value at equilibrium.

$$D = \left( \frac{C_i - C_f}{C_f} \right) * \left( \frac{V_{aq}}{V_{IL}} \right) \tag{5}$$

where $C_i$ and $C_f$ were defined in Equation (1), $V_{aq}$ and $V_{IL}$ are the volume of produced water and IL liquid, respectively.

Equations (6)–(8) are used to calculate the thermodynamics parameters for the removal of oil from PW using IL4. Figure 4b shows the plot of the integrated Van't Hoff equation with an excellent correlation coefficient indicating that ΔH is temperature independent and is equal to 2.0 kJ/mol.

$$lnD = -\frac{\Delta H}{R * T} + C \tag{6}$$

$$\Delta G = -RTlnD \tag{7}$$

$$\Delta G = \Delta H - T\Delta S \tag{8}$$

Equation (7) was used to calculate Gibb's free energy (ΔG) at 298 K with $D$ = 11.12 which gave a value of −5.97 kJ/mol indicating the spontaneity of the extraction process. The change in entropy (ΔS) was calculated using Equation (8) and gave a value of 26.7 J mol$^{-1}$ K$^{-1}$ at 298 K indicating that the mass transfer from the aqueous phase to the IL phase results in an increase in entropy due to freeing water of hydration in the aqueous phase upon transfer to the IL phase.

### 4.10. Effect of Salinity

Different dissolved substances in water can affect the removal of oil from water. The effect of the amount of NaCl on the removal efficiency of IL4 was also studied. For this purpose, solutions of different concentrations of salt were prepared in PW ranging from 250 mg/L to 2000 mg/L. IL4 was tested for each concentration of salt at optimum conditions of initial concentration = 500 mg/L, contact time = 4 min, pH = 8, phase ratio = 4:1 and temperature = 25 °C (Figure 5). Figure 5 reveals that the extraction efficiency decreases with increasing salinity. This decrease in extraction efficiency could be attributed to the interaction of the constituents of oil with the sodium and chloride ions in water which hinders their transfer to the IL phase. Similar behavior has been previously reported for the removal of pollutants from the aqueous phase using ILs [37].

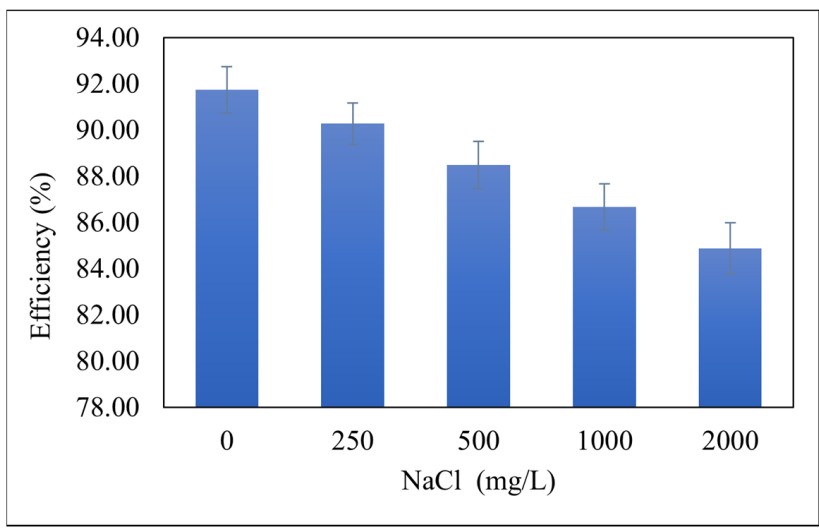

**Figure 5.** Effect of NaCl concentration on the removal efficiency of oil by IL4. Initial oil concentration = 500 mg/L, contact time = 4 min, pH = 8 phase ratio (PW:IL) = 4:1 and temperature = 25 °C, with σ = ±3.37 × 10$^{-1}$%.

### 4.11. Comparison with Other Materials Used for the Produced Water Treatment

Table 4 lists the different adsorbents with their percentage removal efficiency and the time required to remove oil from produced water. Inspection of Table 4 reveals that the reported adsorbents have good efficiency to remove oil from produced water using solid phase extraction, but it takes a longer contact time than that found in this study using liquid–liquid extraction. Our selected ionic liquid also showed good efficiency to remove oil from produced water up to 92.8%. The most efficient IL4 took only 4 min to remove oil from produced water rendering it a valuable extractant for industrial applications.

### 4.12. FTIR Analysis of IL-Oil Interaction

To study the interaction between oil and IL after oil extraction from PW, FTIR spectra were recorded for oil, IL, and IL-oil after extraction (Figure 6). For IL and IL-oil FTIR spectra, the peaks are identified as follows: the peaks in the region of 3500–3600 cm$^{-1}$, 3050–3150 cm$^{-1}$, and 2950–3000 cm$^{-1}$ are due to the presence of a C-H single bond in IL. The peaks in the region of 2000–2100 cm$^{-1}$ are due to the presence of the C≡C bond. The peaks present in the region of 1500–1650 cm$^{-1}$ are due to the presence of C=C and C=N bonds. The peaks present in the region of 600–1500 cm$^{-1}$ are associated with the fingerprint region that is hard to be associated with the complicated existing bonds. The extraction of oil from PW by IL resulted in an observed shift of the peaks of IL for C-H from 3595 to 3600 cm$^{-1}$, and 3030 to 3071 cm$^{-1}$ which represents the successful absorption of oil into IL. The peak for the C≡C bond shifted from 1523 to 1540 cm$^{-1}$ which could be attributed to the incorporation of oil into IL. These shifts in the FITR peaks indicate that oil has successfully been removed from PW and has been absorbed by IL [41,49,50].

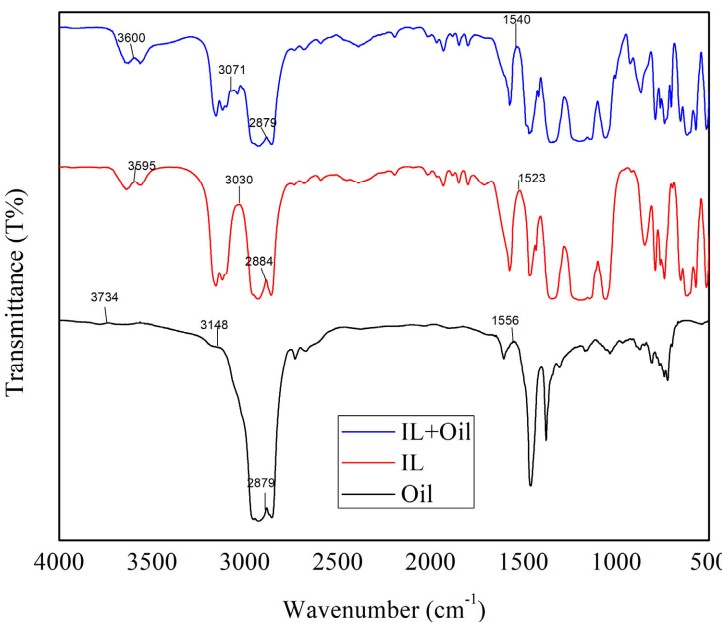

**Figure 6.** FTIR spectra of IL4 before and after extraction of oil from produced water and oil alone.

*4.13. Ionic Liquids Presented Cytotoxic Properties against Human Cell Lines*

Cell viability assays were performed by testing IL4 against human cell lines to determine the metabolic viability of cells as discussed in materials and methods. A confluent cell monolayer was treated with IL4 at different concentrations of human cells. Data are expressed as the mean ± standard error of several experiments performed in duplicates. GraphPad Prism 8.0.2 was used to analyze the data. The overall outcome showed that IL4 presented weak to moderate cytotoxicity against human cell lines, depending on the concentration (Figure 7). Inspection of this figure reveals that IL4 exhibited 36%, 41%, 53%, and 65% cell viability at 2 mM, 1.5 mM, 1.0 mM, and 0.5 mM, respectively. From 0.5 mM, to 1.5 mM, the IL showed high cell viability (Figure 7). It can be concluded that IL4 presented minimal cytotoxic effects at low concentrations and hence can be safely utilized without significant cytotoxicity.

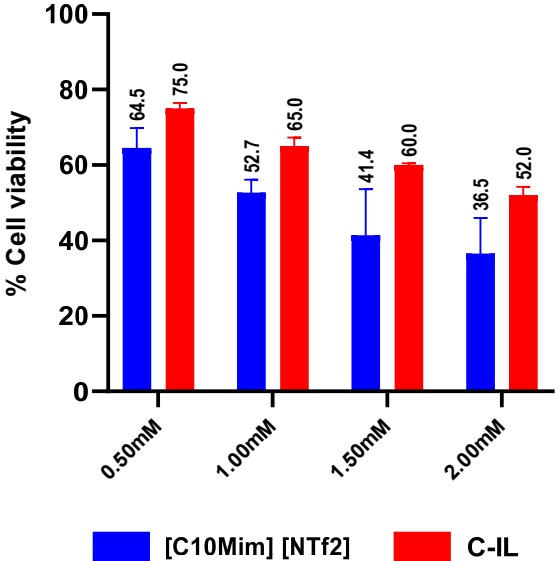

**Figure 7.** The cytotoxic effects of IL at different concentrations on human cells. Blue bars represent NTf2-based ionic liquid and red bars represent a commercial ionic liquid, i.e., IL 1-butyl-1-methylpyrrolidinium bis-(trifluoromethylsulfonyl)imide.

### 4.14. NMR Analysis of IL4

NMR is the most widely used technique to investigate the structure of imidazolium-based ILs [51,52]. It provides a better understanding of the cation and anion interaction. Both $^1$H and $^{13}$C NMR of IL4 spectra were obtained for this purpose. A previous $^1$H NMR study shows that the cation–anion interaction becomes weak with the addition of a new solvent in imidazolium-based IL [53]. Figures 8a and 9a show the $^1$H spectra of IL4 and IL4-Oil, respectively, all the peaks lie in the 0–10 ppm range, where most relevant peaks lie in the 7–10 ppm region belonging to the aromatic hydrogen atoms. After mixing PW with IL, the hydrogen bonding between IL-oil strengthen by the acidic center of cation which causes the chemical shift of peaks for hydrogen atoms (shown in Table 5) and allows the successful removal of oil from PW. A similar $^1$H spectrum has been reported in the literature for the imidazolium-based NTf$_2$ IL [54]. Similarly, Figures 8b and 9b represent the $^{13}$C spectra of the tested IL4 and IL4-Oil, respectively. Inspection of the figure reveals that all the peaks lie in the range of 0–150 ppm region. Upon mixing the PW with IL, the interaction between the cation and anion of IL becomes weak and the stronger interaction between IL-Oil causes the chemical shift in C atoms (shown in Table 5) and allows the mass transfer of oil into IL. The literature reported similar $^{13}$C spectra for the imidazolium NTf$_2$ IL [54]. Table 5 represents the peaks shifts of H and C atoms in IL4 before and after the oil extraction.

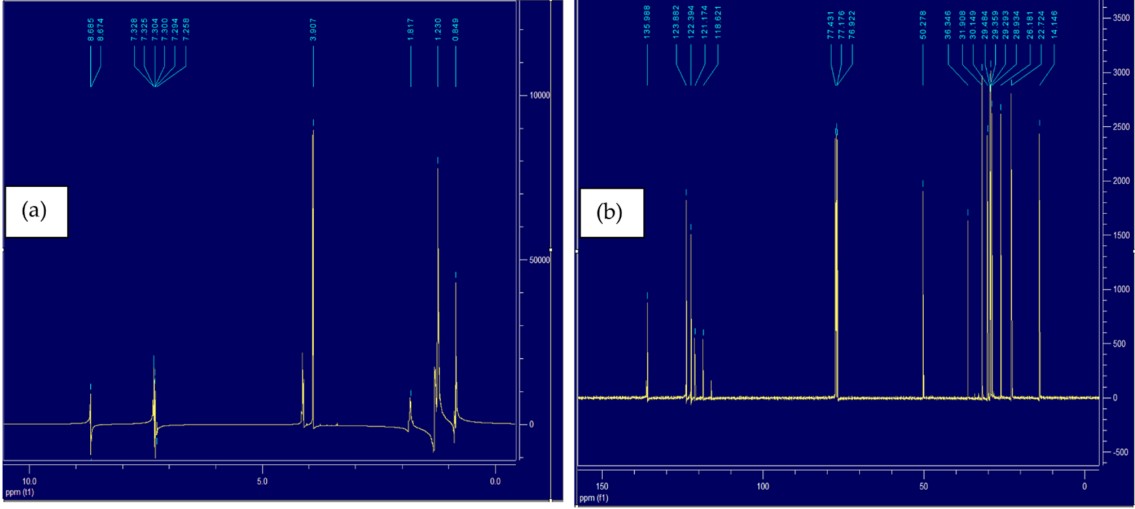

**Figure 8.** (**a**) $^1$H NMR spectra of pure IL4, (**b**) $^{13}$C NMR spectra of pure IL4.

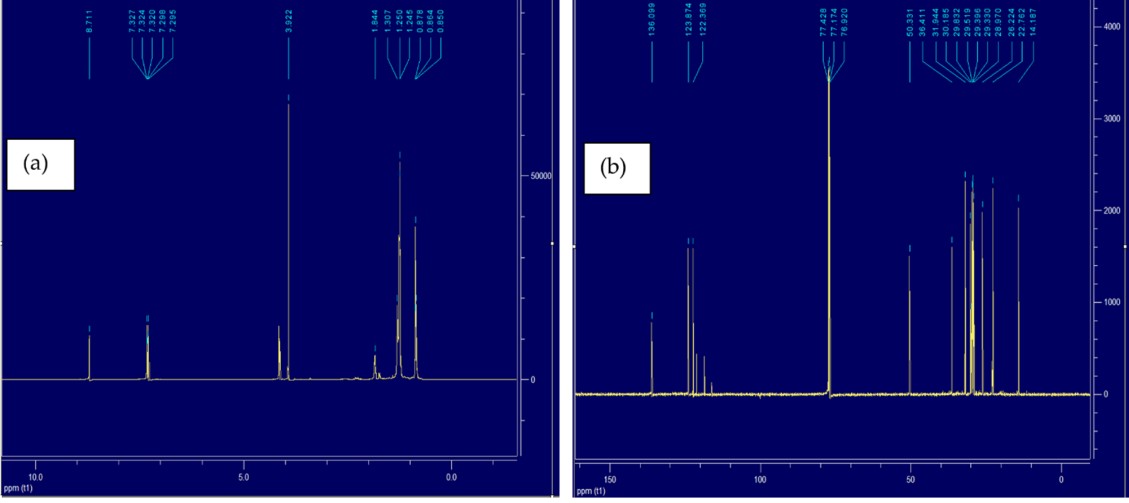

**Figure 9.** (**a**) $^1$H NMR spectra of IL4-Oil, (**b**) $^{13}$C NMR spectra of IL4-Oil.

**Table 5.** NMR peaks shifts for 1H and 13C spectra for IL4 before and after the oil extraction.

| H Peaks Shifts | | C Peaks Shifts | |
|---|---|---|---|
| **IL (ppm)** | **IL4-Oil (ppm)** | **IL (ppm)** | **IL4-OIL (ppm)** |
| 8.685 | 8.711 | 135.988 | 136.099 |
| 7.258 | 7.295 | 123.882 | 123.874 |
| 7.294 | 7.298 | 122.394 | 122.369 |
| 7.300 | 7.320 | 77.431 | 77.428 |
| 7.304 | 7.324 | 50.278 | 50.331 |
| 7.325 | 7.327 | 36.346 | 36.411 |

## 5. Conclusions

PW has adverse effects on human health and aquatic life; therefore, it is essential to remove its oil content before discharging it into the environment. In this study, seven -NTf$_2$-based hydrophobic ionic liquids with different cations: imidazolium, pyridinium, phosphonium, and ammonium, and different alkyl chain lengths: $C_2$, $C_4$, $C_8$ to $C_{10}$, were studied for the removal of oil from produced water. Effects of alkyl chain length and cation nature on extraction efficiency were studied. It was found that the structure of IL affects extraction efficiency. Experimental results indicate that imidazolium-based ionic liquids with higher alkyl chain lengths ([$C_{10}$Mim][NTf$_2$]) showed a maximum efficiency of 92.8% compared to other cations and lower carbon chain lengths. The effects of different process parameters: initial concentration, contact time, pH, phase ratio, temperature, and salt, were also studied. Extraction efficiency increased with increasing concentration and remained almost constant for contact time. It was noted that the extraction is more favorable in the slightly basic region as compared to the acidic region. Extraction efficiency was decreased when the phase ratio of PW:IL was increased. It was observed that temperature did not significantly affect extraction efficiency, therefore, room temperature was selected for the extraction process to save energy and reduce cost. It was noticed that the presence of salts decreased the extraction efficiency. A maximum extraction of (92.8%) was recorded after parameters optimization. Moreover, FTIR and toxicity analyses of the best IL were also performed. Results showed that [$C_{10}$Mim][NTf$_2$] presented higher viability at lower concentrations and was efficient to remove oil from produced water.

**Author Contributions:** S.L. performed the experiments. A.S.K. supervised the experimental work. T.H.I., M.I.K., P.N. and M.Y.A. analyzed the data and reviewed the manuscript. N.A., R.S. and N.A.K. were responsible for the cytotoxicity studies. All authors have read and agreed to the published version of the manuscript.

**Funding:** This research received no external funding.

**Data Availability Statement:** All the data is available in manuscript.

**Acknowledgments:** The authors thank the American University of Sharjah for supporting this work through Research grant FRG19-L-E13.

**Conflicts of Interest:** Authors declare no conflict of interest.

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
