# Peer review of "Hydrophobic Ionic Liquids for Efficient Extraction of Oil from Produced Water"

_processes, doi:10.3390/pr10091897_

Round 1

Reviewer 1 Report

The article submitted by Taleb H. Ibrahim and co-workers under the title 'Hydrophobic Ionic Liquids for Efficient Extraction of Oil from Produced Water' (Manuscript ID: processes-1872462) towards the publication in Processes disclosed the authors new work on the synthesis of hydrophobic ionic liquid, 1-decyl-3-methylimidazolium bis(trifluoromethylsulfonyl)imide ([C10mim][NTf2]) and its application in the extraction of oil from synthesized produced water by a liquid-liquid extraction. The procedures as well as the characterization data presented in this article are logical and the discussions are well-managed. The cell viability tests indicated that the synthesized ionic liquid showed minimal cytotoxic effect at lower concentrations against human cell lines. Besides these the urgency in the development of new systems for the environmental remediation like the removal of oil from contaminated water, this article seems to be qualified to publish in Processes after addressing the following.

1. Authors are suggested to characterize the ionic liquid, IL4 using 1H and 13C NMR, which give better information about its structure.

2. All the references should be formatted to the required style of the journal, 'Processes' and all the list of authors may be provided in all the references. 

3. The initial submission of the article shows several typographical errors: they should be corrected.

4. Introduction may be improved with the discussions related to the enhancement of global population and its related consequences, and urgency of treating waste water. In this connection, the articles Bioresour. Technol. 2022, 346, 126606, Environ. Chem. Lett. 2021, 19, 3887–3950, J. Environ. Manag. 2021, 297, 113316, Bioresour. Technol. Rep. 2022, 18, 101107, J. Environ. Chem. Eng. 2021, 9, 106553 and Green Chem. Lett. Rev., 2021, 14, 700-712 are helpful, which also improve the introduction section.

Reviewer 2 Report

This work presents well-written analysis that describes the aspects of ionic liquids utilization for oil extraction from produced water.

Page 11. Figure 4. Authors claimed that:

“Contact time between PW and IL is an important factor for the liquid-liquid extraction of oil from PW”

How contact time is an important factor and the figure shows no effect on efficiency over time? Why do authors use a contact time of 4 minutes since 2 min shows comparable removal efficiency?

Page 14. Figure 7. Same for temperature. 

Reviewer 3 Report

This manuscript presented oil extraction from produced water using several ionic liquids. The authors performed a systematic oil removal study with various benchmarks. The work has some novelty, but the manuscript has many typos and syntax errors. Please proofread manuscripts before submission. The manuscript cannot be accepted in its current state for publication, and major revisions are suggested. Also, the reviewer has the following specific comments to the authors.

1.      Please replace ppm with mg/L, and “ml” should have a capitalized “L”

2.      How many times were these experiments repeated? Some figures are missing the error estimates; please include the uncertainty in all the figures.

3.      Section 3.2 what is the oil detection limit for the spectroscopy technique?

4.      Section 4.1 – This discussion is very vague and handwaving. Please give examples with relevant Chemistry explanations and strengthen the discussion

5.      The authors need to move the performance comparison table to the manuscript. Also, the table lacks vital information such as the initial concentrations used, experiment conditions etc. Removal percentage has no sense without this information

6.      The manuscript has too many figures, and they may be combined appropriately

7.      All the figures must be B/W print friendly. Ex:- Figure 2 bars can be filled with different patterns etc.

8.      Label the important IR bands on the figure

9.      Figure 7 – the efficiency range can be reduced to 85-95 %, so the slope is visible
